Low-carbohydrate diets differing in carbohydrate restriction improve cardiometabolic and anthropometric markers in healthy adults: A randomised clinical trial

Harvey Cliff J. d. C. 1 cliff@hpn.ac.nz
http://orcid.org/0000-0001-8865-7821 Schofield Grant M. 1
http://orcid.org/0000-0001-6185-7663 Zinn Caryn 1
Thornley Simon J. 1
http://orcid.org/0000-0001-6109-8513 Crofts Catherine 1
Merien Fabrice L. R. 2
1 Human Potential Centre, Auckland University of Technology , Auckland , New Zealand
2 AUT-Roche Diagnostics Laboratory, School of Science, Auckland University of Technology , Auckland , New Zealand
Menini Stefano
Electronic publication date: 2019 Feb 5
Publication date: 2019
Volume: 7
Electronic Location ID: e6273
Received 2018 Oct 2; Accepted 2018 Dec 11
Copyright: © 2019 Harvey et al.
Copyright year: 2019
Copyright holder: Harvey et al.
License: This is an open access article distributed under the terms of the Creative Commons Attribution License, which permits unrestricted use, distribution, reproduction and adaptation in any medium and for any purpose provided that it is properly attributed. For attribution, the original author(s), title, publication source (PeerJ) and either DOI or URL of the article must be cited.
License URL: https://creativecommons.org/licenses/by/4.0/

Keywords: Low-carbohydrate, Diet, Nutrition, Ketogenic, Adherence, Carbohydrate restriction, LCHF, Cardiometabolic health

Funding: Human Potential Centre, Auckland University of Technology All funding was provided by the Human Potential Centre, Auckland University of Technology. The funders had no role in study design, data collection and analysis, decision to publish, or preparation of the manuscript.

==============================
Background

Low-carbohydrate, high-fat (LCHF) diets are useful for treating a range of health conditions, but there is little research evaluating the degree of carbohydrate restriction on outcome measures. This study compares anthropometric and cardiometabolic outcomes between differing carbohydrate-restricted diets.

Objective

Our hypothesis was that moderate carbohydrate restriction is easier to maintain and more effective for improving cardiometabolic health markers than greater restriction.

Design

A total of 77 healthy participants were randomised to a very low-carbohydrate ketogenic diet (VLCKD), low-carbohydrate diet (LCD), or moderate-low carbohydrate diet (MCD), containing 5%, 15% and 25% total energy from carbohydrate, respectively, for 12-weeks. Anthropometric and metabolic health measures were taken at baseline and at 12 weeks. Using ANOVA, both within and between-group outcomes were analysed.

Results

Of 77 participants, 39 (51%) completed the study. In these completers overall, significant reductions in weight and body mass index occurred ((mean change) 3.7 kg/m2; 95% confidence limits (CL): 3.8, 1.8), along with increases in high-density lipoprotein cholesterol, low-density lipoprotein cholesterol, (0.49 mmol/L; 95% CL; 0.06, 0.92; p = 0.03), and total cholesterol concentrations (0.11 mmol/L; 95% CL; 0.00, 0.23; p = 0.05). Triglyceride (TG) levels were reduced by 0.12 mmol/L (95% CL; −0.20, 0.02; p = 0.02). No significant changes occurred between groups. The largest improvements in high density lipoprotein cholesterol (HDL-c) and TG and anthropometric changes occurred for the VLCKD group.

Conclusions

Low-carbohydrate, high-fat diets have a positive effect on markers of health. Adherence to the allocation of carbohydrate was more easily achieved in MCD, and LCD groups compared to VLCKD and there were comparable improvements in weight loss and waist circumference and greater improvements in HDL-c and TG with greater carbohydrate restriction.

Introduction

Low-carbohydrate, high-fat (LCHF) and very low-carbohydrate ketogenic diets (VLCKD) are increasingly used for the management of a range of health conditions, including neurological disorders, obesity, diabetes, metabolic syndrome, and various cancers (Castro et al., 2015; Henderson et al., 2006; Keene, 2006; Kulak & Polotsky, 2013; Lefevre & Aronson, 2000; Levy et al., 2012; Maalouf, Rho & Mattson, 2009; Neal et al., 2008; Paoli et al., 2013; Sumithran & Proietto, 2008; Varshneya et al., 2015). They are also used widely in the general population for weight-loss and maintenance, (Bueno et al., 2013) with improved satiety and control of hunger frequently reported by those who adhere to these diets (Johnstone et al., 2008; McClernon et al., 2007; Paoli et al., 2015). Despite the potential offered by LCHF and low-carbohydrate, high-protein diets, there is little evidence for the superiority of greater carbohydrate restriction compared to moderate. Systematic reviews show that despite greater weight- and fat-loss initially, over longer timeframes, when energy intake is restricted, there is little difference in outcomes for weight-loss, total and low density liproprotein cholesterol (LDL-c) concentrations between diets that are higher or lower in carbohydrate (Hernández Alcantara, Jiménez Cruz & Bacardí Gascón, 2015; Huntriss, Campbell & Bedwell, 2017; Naude et al., 2014; Snorgaard et al., 2017; Van Wyk, Davis & Davies, 2016). However, there are greater reductions in fasted glucose concentrations (Snorgaard et al., 2017), and greater improvements in high density lipoprotein cholesterol (HDL-c) and glycated haemoglobin (HbA1c) with greater degrees of carbohydrate restriction (Huntriss, Campbell & Bedwell, 2017). Controversy exists about the nature of low-carbohydrate diets (LCD) and VLCKDs (Wood & Fernandez, 2009), and definitions for LCDs range from 20 to 200 g of carbohydrate per day (Last & Wilson, 2006; Westman et al., 2007), or up to 40–45% of daily energy from carbohydrate (Hu et al., 2012; Wheeler et al., 2012). Definitions for VLCKDs are similarly vague. The accepted definition for nutritional ketosis (NK) in the clinical nutrition field has become the achievement of ≥0.5 mmol/L ß- hydroxybutyrate, as the majority of people following a VLCKD achieve this level of blood ketones (Gibson et al., 2015), and this threshold has been used by several studies as an indicator of entry into NK (Guerci et al., 2003; Harvey et al., 2018). Ketonaemia consistent with NK typically results from diets containing a 3:1–4:1 ratio of lipids to non-lipid macronutrients, or at least 75% of calories coming from lipids, very low carbohydrates (often less than 50 g) and low-to-moderate amounts of protein (Livingston, Pauli & Pruce, 1977; Livingstone, 1972), or diets containing 60–75% of calories from lipids that include a high proportion of medium chain triglycerides (Huttenlocher, Wilbourn & Signore, 1971; Huttenlocher, 1976). Studies report that adherence is difficult with extreme carbohydrate restriction, that is, <50 g of carbohydrate per day (Huntriss, Campbell & Bedwell, 2017), but insulin-resistant (IR) people may be less likely to adhere to a low-fat, high-carbohydrate diet, compared to those who are more insulin-sensitive (IS). Adherence and weight-loss are similar between both IR and IS participants allocated to a less restrictive LCD (McClain et al., 2013).

Few studies directly compare very LCDs with less extreme carbohydrate-restricted diets. Johnstone and colleagues compared the effects of a non-ketogenic LCD (fat 30% of total energy (TE); carbohydrate 40% of TE) to a ketogenic, LCD (fat 60% TE; carbohydrate 5%TE) in 20 adults over 6 weeks, finding that the diets were equally effective in reducing body weight and insulin resistance (Johnston et al., 2006).

Our hypothesis was that moderate carbohydrate restriction may be easier to maintain, and thus more effective than greater degrees of carbohydrate restriction. The aim of the present study therefore, is to compare anthropometric and cardiometabolic outcomes between a VLCKD, LCD, and moderate-low carbohydrate diet (MCD), containing 5%, 15%, and 25% TE from carbohydrate, respectively, in healthy adults.

Materials and Methods

Population

A total of 77 participants, 25 males, 52 females (mean age: 39 years, range: 25–49; mean body mass index (BMI) 27 kg/m2, range: 20–39) were recruited between the 7th and 19th of August 2017 and gave written, informed consent to participate in this 12-week, randomised, clinical intervention study. The study took place between 11th September and 10th December 2017. Collection of data and analysis was performed at AUT’s Human Potential Centre, Auckland, New Zealand.

Inclusion and exclusion criteria

Participants were required to be healthy and between the ages of 25 and 49 years. Exclusion criteria were; underweight (<18.5 BMI kg/m2), diagnosed with diabetes, diagnosed with any serious medical condition, having previously following a ketogenic diet, or being a current or former client of any of the researchers in clinical practice.

Ethical approval

The trial was registered by the Australia New Zealand Clinical Trial Registry. (ACTRN12617000421336p). Ethics approval for this study was granted by the Southern Committee of the Health and Disability Ethics Committee of New Zealand. 17/STH/60.

Dietary interventions and allocation

Participants completed baseline testing of blood and basic anthropometric measures and a lead-in dietary recording week to identify habitual calorie intake. Participants were randomised by the study statistician to one of three LCD plans which advised intakes of either 5%, 15%, or 25% of TE from carbohydrate. The randomisation was stratified by gender, using a pre-prepared sequence, with investigators blinded to treatment allocation at baseline and follow-up. Participants were assigned to the next treatment group according to their order of recruitment. The primary researcher responsible for initial statistical analysis was blinded to the treatment group allocation until this analysis had been completed.

Diet plans, which included macronutrient and calorie allocation and a sample menu plan, were individualised to the participant, with energy intake determined by the mean reported energy consumed per day in the lead-in dietary recording week. Advice was given to limit protein intake to 1.4 g/kg/day (weight at baseline testing), consistent with International Society of Sports Nutrition guidelines for optimal protein intake for performance (Campbell et al., 2007). This was chosen as an appropriate protein intake that was not likely to unduly influence the study results, because the study participants were healthy people, who may also be engaged in physical activity and sports. Participants were advised to adhere as strictly as possible to the energy and macronutrient prescription for the first 3 weeks of the intervention. For the final 9 weeks of the intervention, they were advised to eat ad libitum but to adhere as closely as possible to the carbohydrate energy limit for their treatment group as a percentage of their TE intake. Usual exercise patterns were continued. Dietary intake was recorded by participants in a mobile application (Fat Secret) with the researchers able to obtain real time entry on a partner mobile application (Fat Secret Pro). Results were monitored for safety and compliance by the primary researcher and research assistants tasked with data-monitoring. Compliance to the dietary allocation was monitored daily by a data monitoring team. Where non-compliance to the dietary allocation, especially for carbohydrate, was noticed, the participant was notified and offered support and advice.

Figure 1 profiles the instructions for the dietary allocations over the 13-week study course.

Figure 1 Flow of participants with dietary allocations during the study period.

Participants were instructed to contact either the clinical nutritionist or the registered dietitian in the research team for any assistance during the study duration.

Anthropometry

The following measures were taken: height, weight, waist circumference at the narrowest point between the lowest rib and the iliac crest, and hip circumference at the widest point of the hips and buttocks. These measures were then used to derive BMI, waist-hip ratio, and the waist-height ratio at baseline and during follow-up.

Blood measures

Following an overnight fast, blood samples were obtained from participants, before the first meal, via venipuncture by a certified phlebotomist from an antecubital vein and collected into plasma separation tubes (PST) Vacutainer tubes using lithium-heparin as the anticoagulant (Becton Dickinson, Franklin Lakes, NJ, USA). Within 15 min of collection, tubes were centrifuged at 1,500×g revolutions per minute for 10 min at +4 °C, and plasma samples were transferred into clean polypropylene tubes and frozen at −80 °C until analyses were conducted using specific diagnostics assays on a Roche Modular analyser (P800 and E170). Blood samples were analysed for total cholesterol (Total-c), LDL-c, HDL-c, triglycerides (TG), C-reactive protein (CRP), gamma-glutamyl transferase (GGT), alanine aminotransferase (ALT), aspartate aminotransferase, alkaline phosphatase (ALP), glucose and uric acid on the P800 module. Insulin, and C-peptide concentrations were measured on the E170 module. All analytical biomarkers were measured at baseline and immediately following the 12-week intervention. The total duration of the assay for each analyte was less than 20 min based on the electrochemiluminescence principle (ruthenium-conjugated monoclonal antibodies) for the E170 module and specific enzyme assay methods for the P800 module. Quantitative results were determined via instrument-specific full point calibration curves and validated with specific controls. Additional information for analytes, lower limits of measurement, measuring range, and test principle can be found in Appendix 1.

Statistical analyses

Effects of the dietary interventions on outcomes were determined for each participant by calculating the change in the various measures from baseline. The significance of these within-group changes from baseline was determined by a paired t-test. All between-group variations were compared using ANOVA. A 5% two-sided alpha level was used to determine significance. Further comparisons were made by undertaking multiple linear regression with adjustment made for variables recorded at baseline. A sensitivity analysis of the results was carried out using stabilised inverse-probability of completing weights for the BMI change outcome to check whether these results were likely to have been different had the whole group returned for followed-up.

Results

A total of 283 people were assessed for eligibility with 206 excluded and 77 included for randomisation to the trial groups (Fig. 2). A total of 10 participants withdrew after they were randomised. Two failed to comply with guidelines to submit baseline data and withdrew from the study (one male, one female), and three females withdrew due to changes in personal circumstances, including two who became pregnant. A further five withdrew due to challenges arising from following the diets. The reasons for withdrawals were as follows: two female participants found the dietary allocation of carbohydrate too difficult to sustain (one each in the 5% and 15% allocation groups); one did not want to continue tracking with the food app; one felt that she could not maintain her sports performance on 15% TE from carbohydrate; and one female in the 5% allocation group reported amenorrhea and reductions in strength and power, despite improved mental clarity. A further 28 did not book for or failed to present for post-intervention measurements. This left 39 participants with follow-up results available for analysis.

Figure 2 Participants included for participation, randomisation, allocation, and lost to follow up.

There were no significant differences in baseline characteristics between completers and non-completers and no meaningful difference in the number of non-completers by group with 50%, 50%, and 48% of participants not completing post-intervention measures in the MCD, LCD, and VLCKD groups respectively. Mean baseline levels of TG were, however, 36% higher at baseline in those lost to follow-up compared to those who were not, even though the difference between the two distributions was not significant (p = 0.08). There was also no significant variation for age, gender, or ethnicity between the groups, in the participants analysed. At baseline, blood measures were all within reference ranges except for Total-c which had an overall mean of 5.31 mmol/L (SD = 1.29) for completers, and a significant between-group difference (p = 0.005).

Baseline characteristics of those included for analysis are presented in Table 1, by randomised treatment group.

Table 1 Baseline characteristics of study participants.

	Treatment group	Total	Test	p-value	
MCD	LCD	VLCKD	
	12	13	14	39			
Age mean (SD)	39.1 (6.6)	38.9 (8.3)	38.7 (7.1)	38.9 (7.1)	ANOVA	0.992	
Gender (%)					Fisher’s	0.198	
 Female	10 (83.3)	6 (46.2)	9 (64.3)	25 (64.1)			
 Male	2 (16.67)	7 (53.85)	5 (35.71)	14 (35.9)			
Ethnicity (%)					Fisher’s	0.733	
 Asian	1 (8.3)	0 (0.0)	1 (7.1)	2 (5.1)			
 European	8 (66.7)	11 (84.6)	10 (71.4)	29 (74.4)			
 Maori	2 (16.7)	1 (7.7)	3 (21.4)	6 (15.4)			
 Other ethnicity	1 (8.3)	0 (0.0)	0 (0.0)	1 (2.6)			
 Pacific peoples	0 (0.0)	1 (7.7)	0 (0.0)	1 (2.6)			
Total energy (Kcal) mean (SD)	1,435 (293)	1,567 (666)	1,805 (857)	1,603 (649)	ANOVA	0.378	
Weight (kg) mean (SD)	76.3 (14.9)	90.4 (20.0)	76.8 (11.2)	81.2 (16.6)	ANOVA	0.046	
Height (m) mean (SD)	1.70 (0.10)	1.76 (0.08)	1.74 (0.09)	1.73 (0.09)	ANOVA	0.245	
BMI (kg/m2) mean (SD)	26.4 (3.23)	29.1 (4.92)	25.5 (2.77)	27.0 (3.96)	ANOVA	0.050	
Glucose (mmol/L) mean (SD)	5.54 (0.43)	5.38 (0.47)	5.44 (0.44)	5.45 (0.44)	ANOVA	0.673	
Total cholesterol (mmol/L) mean (SD)	5.20 (1.3)	4.57 (0.61)	6.10 (1.37)	5.31 (1.29)	ANOVA	0.005	
Triglyceride (mmol/L) mean (SD)	0.79 (0.2)	0.99 (0.36)	0.92 (0.22)	0.90 (0.27)	ANOVA	0.184	
Insulin (pmol/L) mean (SD)	63.1 (37.3)	81.1 (39.4)	41.6 (17.6)	61.4 (35.8)	ANOVA	0.012	
Note:

SD, standard deviation; BMI, body mass index.

Anthropometry

Mean weight and BMI at baseline differed between groups (p = 0.046 and 0.050, respectively). The LCD group had the highest starting BMI at baseline of 29.1 kg/m2 (SD = 4.9), followed by MCD (BMI = 26.4 kg/m2, SD = 3.2). The lowest starting BMI was in the VLCKD group with a mean BMI of 25.5 kg/m2 (SD = 2.8). Overall, there was a significant reduction in weight across all groups (p < 0.001). Mean weight loss increased with the magnitude of carbohydrate restriction, with 4.12 kg (SD = 2.54), 3.93 kg (SD = 3.71), and 2.97 kg (SD = 3.25) lost by the VLCKD, LCD, and MCD groups, respectively. However, the differences in weight loss between these groups were not statistically significant (p = 0.626). Similarly, a highly significant change in BMI of −1.22 kg/m2 (SD = 1.03, p < 0.001) was recorded overall. While the reduction in BMI was greater per magnitude of carbohydrate restriction, this difference was not significant (p = 0.686).

All dietary interventions led to reductions in both waist and hip girth. There was an overall reduction in waist measurement of 2.85 cm (SD = 2.99) and hip girth reduced by 3.43 cm (SD = 4.67, p < 0.001 for both measures). The reduction in waist measurement girth did not differ significantly by group (p = 0.99) but the change in hip girth approached the threshold for significance (p = 0.06). There was a significant change overall to the waist-height ratio (−0.02, p < 0.001) but no significant difference between groups and no significant overall change in the waist-hip ratio. All changes in measures, both overall and by group, with 95% confidence intervals are reported in Table 2.

Table 2 Change in outcome measures, overall, and by group.

Measure	Overall change† Mean change from baseline [95% CI]	Treatment group‡ Mean change from baseline [95% CI]	
Moderate-low carbohydrate diet	Low carbohydrate diet	Very low carbohydrate ketogenic diet	
Weight (kg)	−3.70 [−4.72 to −2.68] p < 0.01	−2.97 [−5.03 to −0.90]	−3.93 [−6.17 to −1.69]	−4.12 [5.58 to −2.65]	
p = 0.63	
Waist circumference (cm)	−2.85 [−3.82 to −1.88] p < 0.01	−2.95 [−5.57 to −0.33]	−2.80 [−4.62 to −0.98]	−2.81 [−3.88 to −1.75]	
p = 0.99	
Hip circumference (cm)	−3.43 [−4.95 to −1.92] p < 0.01	−3.56 [−5.00 to −2.12]	−1.19 [−4.29 to 1.91]	−5.40 [−8.34 to −2.46]	
p = 0.06	
Waist-height ratio	−0.02 [−0.02 to −0.01] p < 0.001	−0.02 [−0.03 to −0.002]	−0.02 [−0.03 to −0.006]	−0.02 [−0.02 to −0.01]	
p = 0.98	
Waist-hip ratio	−0.003 [−0.016 to 0.010] p = 0.66	−0.004 [−0.026 to 0.018]	−0.017 [−0.046 to 0.011]	0.011 [−0.008 to 0.030]	
p = 0.16	
BMI (kg/m2)	−1.223 [−1.556 to −0.889] p < 0.001	−1.031 [−1.757 to −0.306]	−1.22 [−1.894 to −0.546]	−1.39 [−1.899 to −0.881]	
p = 0.686	
Total cholesterol (mmol/L)	0.58 [0.11–1.05] p = 0.02	0.08 [−0.57 to 0.72]	0.94 [0.08–1.80]	0.68 [−0.33 to 1.69]	
p = 0.33	
LDL-c (mmol/L)	0.49 [0.06–0.92] p = 0.03	0.14 [−0.39 to 0.67]	0.80 [−0.02 to 1.62]	0.50 [−0.44 to 1.44]	
p = 0.47	
HDL-c (mmol/L)	0.11 [0.00, 0.23] p = 0.05	−0.05 [−0.33 to 0.24]	0.13 [−0.02 to 0.27]	0.24 [0.07–0.42]	
p = 0.10	
Triglycerides (mmol/L)	−0.12 [0.20 to −0.02] p = 0.02	−0.04 [−0.22 to 0.15]	−0.09 [−0.27 to 0.09]	−0.18 [−0.32 to −0.04]	
p = 0.41	
TG-HDL ratio	−0.101 [−0.173 to −0.030] p = 0.006	−0.023 [−0.123 to 0.078]	−0.118 [−0.294 to 0.058]	−0.154 [−0.259 to −0.048]	
p = 0.31	
Insulin (pmol/L)	−13.58 [−21.61 to −5.56] p < 0.01	−6.45 [−23.38 to 10.48]	−23.68 [−42.49 to −4.86]	−10.33 [−17.03 to −3.62]	
p = 0.19	
Glucose (mmol/L)	−0.11 [−0.26 to 0.04] p = 0.14	−0.22 [−0.55 to 0.11]	0.08 [−0.19 to 0.34]	−0.20 [−0.45 to 0.04]	
p = 0.20	
c-reactive protein (mg/L)	−2.16 [−4.55 to 0.22] p = 0.07	−3.90 [−11.90 to 4.10]	−3.04 [−5.39 to −0.68]	0.14 [−0.50 to 0.77]	
p = 0.34	
Notes:

† Mean change from baseline [95% CI]; p-value relates to repeated measures t-test.

‡ Mean change from baseline [95% CI]; p-value relates to Anova comparing change from baseline within treatment group.

BMI, body mass index; LDL-c, low-density lipoprotein cholesterol; HDL-c, high-density lipoprotein cholesterol.

Blood measures

This paper focuses on the key cardiometabolic outcome measures of Total-c, LDL-c, HDL-c, TG, CRP, glucose, and insulin. Liver enzymes and uric acid were included in the initial analysis as they are emerging markers of interest for metabolic syndrome and insulin resistance (Babio et al., 2015; Ballestri et al., 2016). One participant had GGT levels above the reference range upper limit of 60 U/L. This was reduced from baseline to completion; 143 to 106 U/L. Another participant had baseline levels of ALT of 79 U/L which normalised to 30 U/L at completion (reference range upper limit, 45 U/L). Overall, there was no meaningful change in liver enzymes or uric acid and the differences between groups were not significant.

The most meaningful changes observed were for CRP and insulin. CRP was reduced in the MCD and LCD treatment groups overall by −3.90 mg/L (SD = 12.60), and −3.04 mg/L (SD = 3.90), respectively. There was a marginal increase in CRP in the VLCKD group of 0.14 mg/L (SD = 1.10) which we would not consider to be meaningful. While the overall change from baseline CRP approached the threshold for significance (p = 0.074), there was no difference between the groups (p = 0.339). While at baseline, no significant difference for CRP was present between groups (p = 0.346), there were several readings for CRP that were above the reference range upper limit of five mg/L. The highest reading of 46.9 mg/L was recorded in the MCD group and there were also three readings >5 mg/L in the LCD group, with the highest maximal reading of 13 mg/L. Conversely, the maximal recorded value for CRP in the VLCKD group at baseline was 2.6 mg/L. On follow-up, all results were <5 mg/L.

Insulin concentration was reduced overall by 13.6 pmol/L (SD = 24.8, p < 0.001). The greatest change occurred in the LCD group, followed by the VLCKD group, with the smallest change in the MLC group. The difference between groups, however, was not statistically significant (p = 0.185).

Statistically significant changes, albeit of a relatively small magnitude, occurred for Total-c, LDL-c, and HDL-c, which were all increased at completion vs baseline, and for TG which were reduced, with no significant variation between groups. No meaningful change from baseline was observed for fasted glucose. There was however, a significant improvement in the TG-HDL ratio of −0.102 (SD = 0.220, p = 0.006). This improvement was increased with greater carbohydrate restriction with changes of −0.023 (SD = 0.158), −0.118 (SD = 0.291) and −0.154 (SD = 0.182), for MCD, LCD, and VLCKD, respectively (p = 0.308).

Large proportional changes from baseline occurred for insulin, TG, Total-c, LDL-c, and HDL-c. Proportional increases from baseline for Total-c and LDL-c were greatest for LCD, followed by VLCKD, and MCD. There was no relative change from baseline for both TG and HDL-c in the MCD group. Improvements in HDL-c and TG occurred for the LCD group, with the greatest proportional change in the VLCKD group. There were relatively minor proportional changes for the remaining measures. (Fig. 3.) All changes in reported measures, overall and by group, with 95% confidence intervals, are reported in Table 2.

Figure 3 Percentage change from baseline in cardiometabolic and athropometric outcome measures.

Adherence to diet

The individual mean daily energy intake per group, by week, is shown in Fig. 4. A marginal increase in reported energy intake occurred during the first 3 weeks, during which time participants had been advised to maintain usual energy intake. After the first 3 weeks, participants had been advised to eat ad libitum but preserve the carbohydrate allocation as a percentage of TE intake. In this phase, the pattern of increased energy over baseline was maintained over most weeks but eventually declined. By week 12 there was an overall reduction in mean energy compared to baseline of 66, 95, and 192 Kcal for MCD, LCD, and VLCKD, respectively. So, overall there was a greater magnitude of energy increase initially with greater carbohydrate restriction, but over time this resulted in a greater reduction in TE consumed commensurate with the magnitude of carbohydrate restriction. These changes from baseline were relatively small with the greatest magnitude of change from baseline, 12%, 10%, and 18% for MCD, LCD, and VLCKD, respectively.

Figure 4 Mean daily energy intake by week per participant.

The blue line indicates the linear trend. Black line indicates the 50th percentile.

Over 12 weeks carbohydrate intake by group was less than allocation for both MCD (22.5%, SD = 4.5%) and LCD (14.1%, SD = 3.2%) and higher than allocation for VLCKD (7.9%, SD = 4.9%). A linear trend was observed for reduction in carbohydrate intake as a proportion of TE for MCD relative to week (β = −0.137, p = 0.24). Conversely increased intake by week was observed for LCD (β = 0.096, p = 0.24), and VLCKD (β = 0.174, p = 0.15) but these trends were not significant within groups, or between group allocations (p = 0.108). Figure 5 shows the reported energy per participant, derived from carbohydrate per group, by week.

Figure 5 Mean percent of total energy derived from carbohydrate by participant, per week

The black line indicates the 50th percentile.

Protein intake did not differ between the groups at baseline (p = 0.299). There was no significant variation between groups for average protein intake per day over the course of the study. Fat intake varied by group but was consistent with their TE intake, and protein and carbohydrate allocations.

Discussion

Principal findings

Overall, the results demonstrated that reduced carbohydrate diets have a positive effect on select markers of health. Despite a high number of participants who did not present for follow-up testing, in those included for analysis, LCDs were easily adhered to over a 12-week period. While there was little difference in the consistency of adherence between the differing dietary interventions for calorie and macronutrient allocations overall, carbohydrate intake was more easily maintained in the MCD and LCD groups, as demonstrated by mean intakes lower than allocation, whereas mean intake of carbohydrate as a percentage of TE was higher than allocation in the VLCKD group. There was a marginal increase in energy intake from baseline, but this declined over the course of the study in all groups. Of interest was the relatively low-calorie intake recorded at baseline which might indicate a cohort focussed on weight loss or under-reporting of actual food intake.

Almost all participants began the study with anthropometric and blood measurements within the normal range. We would, therefore, not expect large changes for markers of health in a generally ‘healthy’ cohort. This was also a eucaloric intervention, designed to match habitual energy intake and was not designed as a ‘weight loss’ trial. Despite this, there were significant and clinically meaningful, albeit relatively small, improvements in weight, waist-height ratio, HDL-c, and TG. Of the changes in outcome measures that reached the threshold for significance, seven of nine were improved from baseline favourably (HDL, TG, insulin, weight, waist, hip, and BMI) while only Total-c and LDL-c increased by a small magnitude. Of particular interest, was the improvement in waist-height ratio, as this is a strong predictor of all-cause mortality. (Ashwell et al., 2014) We would also consider the significant improvements in HDL-c and TG to be clinically meaningful measures of interest when compared to relatively minor changes in Total-c or LDL-c. Of all the commonly measured biomarkers of cardiovascular risk, TG concentrations are most convincingly linked to incident cardiovascular disease (Harcombe et al., 2015; Liu et al., 2013; Ravnskov et al., 2016). Reductions in relative risk are seen at TG <1.02 mmol/L, with every one mmol/L increase associated with a >12% increase in risk, for both cardiovascular disease mortality and all-cause mortality (Liu et al., 2013). Interestingly, in the current study, while mean TG levels were reduced in all groups at 12-weeks, only the VLCKD group showed an improvement in TG levels, with a reduction of 0.04 mmol/L at the upper limit of the 95% confidence intervals, compared to an increase of 0.16 and 0.09 mmol/L for the MCD and LCD groups, respectively. This suggests that the higher the baseline TG, the greater the benefit of carbohydrate restriction. Our weighted regression re-analysis also showed that baseline TG affected the change in BMI relative to treatment group, suggesting the hypothesis that baseline lipids may predict outcomes from diets differing in carbohydrate allocation. This hypothesis will be investigated and reported in a separate paper.

There is debate around the respective roles that Total-c, LDL-c, HDL-c, TG, and their interactions play with respect to mortality and morbidity outcomes. This warrants further investigation, especially in the context of reduced carbohydrate diets.

An additional sensitivity analysis was subsequently carried out which modelled the probability of completing the study, given baseline values of age, gender, weight, TG, and glucose concentration using a logistic regression model. These values were then used in a re-analysis of the change in BMI at the end of the study with observations re-weighted by stabilised-inverse probability of treatment from the logistic model. This model showed a larger decrease in mean BMI, comparing the VLCKD to the MCD group (mean change from baseline: −0.59 kg/m2; 95% CI [0.21 to −1.39]). This difference from the unweighted analysis is likely to be due to different effects of the diets by baseline TG concentration. These changes will be explored further in a future analysis.

Several CRP readings were above the reference range of <5 mg/L. The highest reading of 46.9 mg/L, recorded in the MCD group, was found,on subsequent investigation to be due to an unreported flu-like viral infection. At the conclusion of the study, all results for CRP were <5 mg/L. This suggests a positive effect on systemic inflammation from LCDs overall, but high baseline results may have been due to undisclosed illness or another stressor.

Strengths and weaknesses of the study

This study is one of the first to compare eucaloric diets differing in the magnitude of carbohydrate restriction for anthropometric and cardiometabolic outcomes in healthy people. It was a randomised trial, including food tracking with real-time researcher monitoring and feedback, along with advice and information provided to participants from a competent team with extensive experience in the prescription of LCDs and VLCKDs. As such, we believe it provides a valuable addition to the literature to help inform clinical practice.

Our study was limited by small sample size and by the failure of 49% of participants to either complete the intervention or present for follow-up testing. This was expected, as high dropout rates are common in dietary studies. For example, a systematic review of LCDs vs low-fat, calorie restricted diet interventions showed an overall attrition rate of 36%, with a higher rate of attrition in low-fat, high-carbohydrate interventions (Hession et al., 2009). Few participants reported dropping out due to challenges with the diets and most dropouts were instead due to failure to present for testing rather than failure to adhere to the diet, and these numbers were almost identical between the intervention groups. Participants who failed to present were asked to provide reasons for (not) doing so. Two participants responded, stating a clash with work and inability to attend due to parental responsibilities. It is therefore unclear whether there were other factors, outside of scheduling or other logistical challenges, that affected participants completing the study.

The final numbers included in our analysis due to attrition, therefore lacked statistical power. With larger numbers, greater statistical significance may be detected. This will be of value to elucidate the impact of differing magnitudes of carbohydrate restriction on important markers of cardiometabolic health in which there was a between group difference in change from baseline, for example, TG and HDL-c. The small sample size also highlights a potential problem of applying parametric tests, that is, whether or not the data collected fit the probability distributions associated with them. An alternative that does not rely on such assumptions is a randomisation test. Results from these tests in our study were very similar to those obtained from t-tests, for example, the p-value for the between group differences in change from baseline Total-c was p = 0.658 which was very similar to the results of the ANOVA, p = 0.686.

The study also did not include a group with a higher carbohydrate allocation consistent with existing dietary guidelines of 45–75% of energy derived from carbohydrate (Buyken et al., 2018), (i.e. a true control group) and therefore, we cannot discount that higher-carbohydrate, lower-fat diets with an emphasis on high quality food intake, a reduced preponderance of refined, energy-dense foods, nutrition counselling as available in this study, and the accountability of being involved in a study, could lead to similar beneficial results. In the recent dietfits study a higher- and lower-carbohydrate intervention, with nutritional counselling and an emphasis on ‘quality’ nutrition resulted in similar results for weight-loss over 12 months (Gardner et al., 2018). However, in this study there was a non-significant trend towards greater weight loss, and statistically significant improvements in HDL-c and TG in the lower-carbohydrate group. In the present study, these were improved in a dose-dependent fashion per carbohydrate restriction. There is already a large body of evidence comparing low- to high-carbohydrate diets, and this study helps to instead differentiate between differing lower-carbohydrate diets and their benefits.

Meanings and implications of the study

The consistency of the improvements in important predictors of mortality suggest a beneficial effect of lower carbohydrate interventions overall, and similarly, towards greater improvement on the most meaningful markers of health, concomitant to the magnitude of carbohydrate restriction. This is of particular interest because the dietary interventions were not hypocaloric and were designed to match habitual energy intake. Yet despite matching the calorie intake at baseline to the dietary prescription, meaningful anthropometric and blood measures of cardiometabolic health, were improved and trended towards greater (non-significant) improvements with greater carbohydrate restriction. However, the adherence to the carbohydrate allocation was more likely to be achieved in those on more moderate carbohydrate-restricted diets.

Unanswered questions and directions for future research

This study shows positive effects overall from reduced carbohydrate diets on select markers of health and further suggests a potential benefit from a greater magnitude of carbohydrate restriction, despite this greater carbohydrate restriction being more difficult to achieve. Additional research with larger sample sizes is warranted to investigate this further. Due to the large numbers that failed to present for follow-up testing, further investigation is warranted to ascertain factors associated with adherence to the diet.

Conclusion

Low-carbohydrate diets are beneficial for the improvement of anthropometric and blood markers of cardiometabolic health in healthy adults and are easily adhered to over 12-weeks. However, the greatest restriction of carbohydrate to 5% of TE may not be realistically achievable for this population. Our results demonstrate that non-hypocaloric, LCDs, matched to habitual calorie intake, result in significant improvements in predictors of long-term health including weight, waist and hip girth, waist-to-height ratio, TG, and HDL-c, which increase in magnitude with a greater degree of carbohydrate restriction. However, between-group differences typically did not reach thresholds for statistical significance, and further research with larger samples is required to investigate further, the effects of different degrees of carbohydrate restrictions on outcomes in healthy populations.

Appendix 1 Assay Performance

Analyte	Lower limit of measurement*	Measuring range	Test principle	
Total cholesterol	0.1 mmol/L	0.1–20.7 mmol/L	Enzymatic colorimetric test	
LDL-c	0.078 mmol/L	0.078–14.2 mmol/L	Homogeneous enzymatic colorimetric assay	
HDL-c	0.08 mmol/L	0.08–3.10 mmol/L	Homogeneous enzymatic colorimetric assay	
TG	0.05 mmol/L	0.05–11.3 mmol/L	Enzymatic colorimetric test	
CRP	2.9 nmol/L	2.9–3333 nmol/L	Particle-enhanced immunoturbidimetric assay	
GGT	3 U/L	3–1,200 U/L	Enzymatic colorimetric test	
ALT	5 U/L	5–700 U/L	Enzymatic colorimetric test	
AST	5 U/L	5–700 U/L	Enzymatic colorimetric test	
ALP	3 U/L	3–1,200 U/L	Enzymatic colorimetric test	
Glucose	0.11 mmol/L	0.11–41.6 mmol/L	Enzymatic colorimetric test	
Uric acid	11.9 μmol/L	11.9–1,487 μmol/L	Enzymatic colorimetric test	
Insulin	1.39 pmol/L	1.39–6,945 pmol/L	Electrochemoluminescence	
C-peptide	0.003 mmol/L	0.003–13.3 nmol/L	Electrochemoluminescence	
Note:

* Functional sensitivity. It represents the lowest measurable analyte level that can be distinguished from zero. It is calculated as the value lying two or three standard deviations above that of the lowest standard. Method comparisons, limitations, and specific performance data can be found on www.e-labdoc.roche.com.

Supplemental Information

Supplemental Information 1 R data file showing baseline and completion and change from baseline for outcome measures described in this paper.

Click here for additional data file.

Supplemental Information 2 CONSORT checklist for the study presented here.

Click here for additional data file.

Supplemental Information 3 Study protocol.

Click here for additional data file.

We are grateful to the participants in this study. We acknowledge the support of our colleagues at the Human Potential Centre, AUT University, especially Dee Holdsworth-Perks; and at the Holistic Performance Institute. Specifically, we would like to acknowledge the assistance of Amberleigh Jack, Bella Marinkovich, Emily White, and Lulu Caitcheon who assisted with data collection; phlebotomists, Brenda Costa-Scorse and Claudia Barclay; Kirsten Beynon for lab assistance during data collection and analysis, Dave Shaw for his co-operation during data-collection, and Eric Helms and Nigel Harris for advice about the study design and methods.

Additional Information and Declarations

Competing Interests

Author Contributions

Human Ethics

Data Availability

Clinical Trial Registration

The authors declare that they have no competing interests.

Cliff J. d. C. Harvey conceived and designed the experiments, performed the experiments, analysed the data, prepared figures and/or tables, authored or reviewed drafts of the paper, approved the final draft.

Grant M. Schofield conceived and designed the experiments, prepared figures and/or tables, authored or reviewed drafts of the paper, approved the final draft.

Caryn Zinn conceived and designed the experiments, prepared figures and/or tables, authored or reviewed drafts of the paper, approved the final draft.

Simon J. Thornley conceived and designed the experiments, analysed the data, prepared figures and/or tables, authored or reviewed drafts of the paper, approved the final draft.

Catherine Crofts conceived and designed the experiments, performed the experiments, authored or reviewed drafts of the paper, approved the final draft.

Fabrice L. R. Merien performed the experiments, analysed the data, contributed reagents/materials/analysis tools, authored or reviewed drafts of the paper, approved the final draft.

The following information was supplied relating to ethical approvals (i.e. approving body and any reference numbers):

Ethics approval for this study was granted by the Southern Committee of the Health and Disability Ethics Committee of New Zealand (17/STH/60).

The following information was supplied regarding data availability:

Raw data for baseline and outcome measures is available in the Supplementary Dataset.

The following information was supplied regarding Clinical Trial registration:

The trial was registered by the Australia New Zealand Clinical Trial Registry. (ACTRN12617000421336p).

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
