# Peer review of "Low-carbohydrate diets differing in carbohydrate restriction improve cardiometabolic and anthropometric markers in healthy adults: A randomised clinical trial"

_PeerJ, doi:10.7717/peerj.6273_

## Round 0.1 · original submission · Major Revisions

Dear Dr. Harvey,

Thank you for submitting your manuscript entitled " Low-carbohydrate diets differing in carbohydrate restriction improve cardiometabolic and anthropometric markers in healthy adults: A randomised clinical trial" for consideration by PeerJ. I have now received reports from two reviewers who found the paper to be well written and the findings to be of interest, but they have raised concerns that preclude acceptance of the manuscript in its current form. Accordingly, I invite you to respond to the reviewers' comments and recommendations below. The revised version will be re-reviewed and a decision on acceptability of your manuscript will be made only after the revised version has been re-evaluated.

If you decide to resubmit the revised version, please summarize all the improvements made in the new version and give answers to all critical points raised in the reviewers’ report in an accompanying letter. You can find further comments of Reviewer #1 in attachment. Please also consider these comments in your response to reviewers.

I suggest you pay particular attention to the following reviewers’ criticisms/suggestions:

- The statistical analysis section needs major attention. Please, address the criticism and perform more robust and detailed statistical analysis in order to reduce the effects of potential confounders;
- Improve the clarity of concepts, information about the validity and accuracy of the measurement techniques and about the study design.

Sincerely,

Stefano Menini

Reviewer 1 ·

Basic reporting

The basic language throughout is clear, professional and easy to understand. The figures and tables are reported well.

Experimental design

The experimental design was well done. Some of the reporting however is not clear. The statistical analysis is not appropriate.

Validity of the findings

The hypothesis of the researchers was that moderate carbohydrate restriction may be easier to maintain than severe carbohydrate restriction. However, this hypothesis was not objectively measured nor was it reported in an objective manner. The statistical analysis was not done correctly--since there are multiple diet groups, interaction effects, as well as main effects need to be tested and reported. I can't recommend this paper be considered for publication until the statistical analysis is revised and with two way ANOVAs and interaction effects as well as treatment (main effects) are reported clearly.

Additional comments

I believe this is a good study and a great research question. I believe overall, the study was well executed. However, significant modification needs to be made in regards to the statistical analysis and reporting aspect of the paper.

Annotated reviews are not available for download in order to protect the identity of reviewers who chose to remain anonymous.

·

Basic reporting

The language used throughout was appropriate. The cited literature was also appropriate with the main papers in the are having been referred to. The structure was appropriate.

I enjoyed reading the paper but felt the context was lost a little when discussing adherence. The issues with adherence with low carb diets generally come after 1 year so this would be difficult to detect in a 12-week study. The more interesting data, I feel would be from the clinical outcomes. I think you can keep the adherence data and discuss it but not really have this as a standalone/focal hypothesis because it isn’t really anything that your study could conclude on? Also, this objective/hypothesis isn’t mentioned in the title or the background in the abstract, so it feels a little misplaced.

I feel useful data to include would be the mean macronutrient breakdown of each of the groups at baseline and at week 12. Could you include this?

Experimental design

Well defined research question. Good explanation of blinding and how participants were assigned to groups. Could a little more information be given regarding the dietary instruction that participants were given? Lack of power/small sample size an issue but acknowledged in limitations to the study. Ethical approval acquired.

It may read better to have inclusion and exclusion criteria instead of “not diagnosed with diabetes” and “not previously following a keto diet” in inclusion, this could be exclusion criteria.

Validity of the findings

Conclusions seem sound but should say LCHF diets are easily adhered to “over a 12 week period”. It would be useful in the text and in the abstract to give the individual results of each of the 3 groups for each of the outcomes as opposed to the combined results as the combined results isn’t truly sharing the results of a RCT.

The study talks about adherence, but actually the adherence discussed is related to the adherence to calories, whereas the adherence should be discussed in relation to the carbohydrate as that is what we are discussing so the way in which this is reported needs changing.

Additional comments

I enjoyed reading the paper so thank you for the submission. I feel a few areas need to be addressed as per my comments but would make a good paper if these suggestions were actioned.

---

## Round 0.2 · Minor Revisions

Dear Dr. Harvey,

Your manuscript entitled "Low-carbohydrate diets differing in carbohydrate restriction improve cardiometabolic and anthropometric markers in healthy adults: A randomised clinical trial" has again been carefully reviewed by the Editor and Reviewer 2. Basically the revision is now acceptable for publication, but before final acceptance is given, I would appreciate it if you would address the remaining issue on inclusion criteria raised by Reviewer 2.

If you are willing to do this, it would not be necessary for me to return the manuscript to the reviewers, but it could then be accepted for publication.

Sincerely yours,

Stefano Menini

·

Basic reporting

Much better. I felt the authors listened to the comments provided. The objectives were much clearer and therefore the structure and flow of the paper felt much improved.

Experimental design

Again, much better. Thank you for adding this detail.
One small comment, in inclusion criteria to add the age of participants. e.g. > 18 years old.

Validity of the findings

No further comments. The conclusions are sound.

Additional comments

Well done to the authors, I feel the paper reads much better. Thank you for taking the time to make the amendments.

---

## Round 0.3 · accepted · Accept

Dear Dr. Harvey,

I am pleased to inform you that the revision of your manuscript entitled "Low-carbohydrate diets differing in carbohydrate restriction improve cardiometabolic and anthropometric markers in healthy adults: A randomised clinical trial" now makes it acceptable for publication in PeerJ. I appreciate very much your making the suggested revisions.

Sincerely yours,

Stefano Menini

#